# A Critical Review of the Psychomotor Agitation Treatment in Youth

**DOI:** 10.3390/life13020293

**Published:** 2023-01-20

**Authors:** Beniamino Tripodi, Irene Matarese, Manuel Glauco Carbone

**Affiliations:** 1Department of Mental Health and Addictions, Division of Psychiatry, ASST Crema, Via Largo Ugo Dossena 2, 26013 Crema, Italy; 2Pisa-School of Experimental and Clinical Psychiatry, University of Pisa, Via Roma 57, 56100 Pisa, Italy; 3Department of Medicine and Surgery, Division of Psychiatry, University of Insubria, Viale Luigi Borri 57, 21100 Varese, Italy; 4Department of Mental Health and Addictions, Clinical Psychology Service, ASST Crema, Via Largo Ugo Dossena 2, 26013 Crema, Italy; 5Department of Surgical, Medical, Molecular and Critical Area Pathology, Clinical and Health Psychology, University of Pisa, Via Roma 57, 56100 Pisa, Italy

**Keywords:** psychomotor agitation, childhood, children, adolescents, aggressiveness, psychopharmacological treatments

## Abstract

(1) Background: To systematically review evidence on the safety and efficacy of psychopharmacological treatments available for psychomotor agitation (PA) in children and adolescents. (2) Methods: Studies assessing the safety and efficacy of psychopharmacological treatments for acute PA in children and adolescents that were published between January 1984 and June 2022 on PubMed were systematically reviewed. We included: (i) papers that presented a combination of the search terms specified in the “*Search strategy*” sub-paragraph; (ii) manuscripts in English; (iii) original papers; (iv) prospective or retrospective/observational studies and experimental or quasi-experimental reports. The exclusion criteria were: (i) review papers; (ii) non-original studies including editorials and book reviews; (iii) studies not specifically designed and focused on the selected topic. (3) Results: We selected 42 papers: 11 case series (11/42, 26.19%), 8 chart reviews (8/42, 19.05%), 8 case reports (8/42, 19.05%), 6 double-blind placebo-controlled randomized studies (6/42, 14.29%), 4 double-blind controlled randomized studies (4/42, 9.52%), 4 open-label trials (4/42, 9.52%) and 1 case control (1/42, 2.38%). (4) Conclusions: The drugs most frequently used to treat agitation in children and adolescents were ziprasidone, risperidone, aripiprazole, olanzapine and valproic acid. Further studies are needed to evaluate the efficacy/safety ratio, considering the limited number of observations in this field.

## 1. Introduction

### 1.1. Rationale

Psychomotor agitation (PA) endangers the safety of both patients and community healthcare service operators [1]. PA is a state of disorganized and aimless psychomotor activity stemming from an acute condition of physical and mental discomfort. Restlessness with excessive or partially purposeful motor activity, severe irritability with impulse dyscontrol, paradoxical responsiveness to internal and external stimuli, loud and angry speech, muscle tension, autonomic hyper-arousal, diaphoresis, tachycardia and hostility are typical manifestations of PA [2]. Recently, PA received attention in children and adolescents considering that the prevalence of pediatric diagnoses with PA episodes, such as mood disorders (more frequently Bipolar Disorder [BD] than Major Depressive Disorder), psychotic disorders, impulse control and conduct disorders, Substance Use Disorder, Post-Traumatic Stress Disorder and neurodevelopmental disorders (particularly Autism Spectrum Disorder and/or Intellectual Disability), has increased 40-fold between 1994 and 2003 in outpatient settings [3,4]. However, the psychotropic medications commonly used for PA are, for the vast majority of cases, not formally indicated to treat children and adolescents, except for specific medications for specific indications, which differ in different countries according to their regulatory rules (i.e., mood disorders, psychotic disorders, behavior disorders and irritability in autism spectrum disorders or intellectual disability, etc.). This led to the paradox of a ‘*usual*’ condition of off-label prescriptions for the younger agitated patients [5,6]. Two main attempts have been made to define evidence-based guidelines for PA in children and adolescents. The ‘*Center for the Advancement of Children*’*s Mental Health*’ at Columbia University and the ‘*New York State Office of Mental Health*’ outlined two evidence-based and consensus-based treatment recommendations for the use of antipsychotics for aggressive youth (TRAAY) [7,8]. In summary, they stated that there was only ‘*empirical evidence*’ regarding the efficacy of pharmacological and non-pharmacological treatments of PA in younger people. Most of the controlled research on atypical antipsychotics was on risperidone, which is effective and has less extrapyramidal side effects than typical antipsychotics. Clozapine, olanzapine, β-blockers, the α-agonist clonidine and stimulants (the last being only for ADHD-related PA) were considered as alternative treatments. Moreover, recommendations regarding the different steps (such as ‘*evaluation*’, ‘*treatment*’, ‘*stabilization*’ and ‘*maintenance*’) were defined. In 2015, the ‘*National Collaborating Center for Mental Health*’ (NCCMH), commissioned by the ‘*National Institute for Health and Care Excellence*’ (NICE), released an update of ‘*Violence: the Short-term Management of Disturbed/Violent Behaviors in inpatient psychiatric Settings and Emergency Departments*’, a guideline developed by the ‘*National Collaborating Center for Nursing and Supportive Care*’ which was originally published in 2005 with the aim of providing help to clinicians in the short-term management of violence and aggression in adults, children (aged 12 years or under) or young people (aged 13 to 17 years) [9]. Specifically, the section 7 of this guideline was dedicated to the treatment of agitation in children and adolescents. As a first step, it is advised to proceed with de-escalation techniques. In cases of failure, the suggestion was “*to contain agitation with restrictive and pharmacological interventions*”. Intramuscular lorazepam is recommended, adjusting the dose to the young person’s age and weight. Despite all these guidelines and recommendations, the evidence-based findings for pharmacological options appear to be limited or anecdotal.

### 1.2. Objectives

The main aim of this paper is to systematically review and summarize the available findings from studies on the safety and efficacy of psychopharmacological options for PA in children and adolescents. We set out to systematically review the published literature on the topic in accordance with the PICOS process as follows: P—population: female and male patients < 18 years of age who met the clinical criteria for PA; I—intervention: psychopharmacological treatment for children and adolescents with PA; C—comparison: patients with PA before and after psychopharmacological treatment and matched groups treated with other forms of treatment (when available) or control groups (when available); O—outcome: changes in the severity of PA symptoms or the number of episodes of PA, however expressed (absolute value, z- or t-scores standard deviations, increases in percentage from baseline to follow-up); S—study design: we included randomized controlled trials (RCTs), cohort studies, case-control studies, follow-up studies, pilot studies, quasi-experimental studies, case series or case reports.

## 2. Materials and Methods

We adhered to the Preferred Reporting Items for Systematic Review and Meta-Analyses (PRISMA) guidelines in the completion of this systematic review [10].

### 2.1. Protocol and Registration

This systematic review was not included in a research protocol.

### 2.2. Eligibility Criteria/Information Source

All studies published between January 1984 and June 2022 using PubMed were included, provided that they met the following criteria: (1) written in English; (2) original articles on studies with a longitudinal design; (3) prospective or retrospective, observational (analytical or descriptive), experimental or quasi-experimental, controlled or non-controlled studies; (4) case series, case reports and letters to the editor.

### 2.3. Search Strategy

We searched eligible literature for this systematic review through PubMed up to June 2022, with the following search terms:

(1) “(agitation[Title]) AND pediatric[Title]”; (2) “(agitation[Title]) AND children[Title]”, (3) “(agitation[Title]) AND adolescent[Title]”; (4) “(olanzapine[Title]) AND psychomotor agitation[Title] AND children[Title]”; (5) “(olanzapine[Title]) AND psychomotor agitation[Title] AND adolescents[Title]”; (6) “(clozapine[Title]) AND psychomotor agitation[Title] AND children[Title]”; (7) “(clozapine[Title]) AND psychomotor agitation[Title] AND adolescents[Title]”; (8) “(quetiapine[Title]) AND psychomotor agitation[Title] AND children[Title]”; (9) “(quetiapine[Title]) AND psychomotor agitation[Title] AND adolescents[Title]”; (10) “(aripiprazole[Title]) AND psychomotor agitation[Title] AND children[Title]”; (11) “(aripiprazole[Title]) AND psychomotor agitation[Title] AND adolescents[Title]”; (12) “(paliperidone[Title]) AND psychomotor agitation[Title] AND children[Title]”; (13) “(paliperidone[Title]) AND psychomotor agitation[Title] AND adolescents[Title]”; (14) “(ziprasidone[Title]) AND psychomotor agitation[Title] AND children[Title]”; (15) “(ziprasidone[Title]) AND psychomotor agitation[Title] AND adolescents[Title]”; (16) “(haloperidol[Title]) AND psychomotor agitation[Title] AND children[Title]”; (17) “(haloperidol[Title]) AND psychomotor agitation[Title] AND adolescents[Title]”; (18) “(benzodiazepines[Title]) AND psychomotor agitation[Title] AND children[Title]”; (19) “(benzodiazepines[Title]) AND psychomotor agitation[Title] AND adolescents[Title]”; (20) “(lithium[Title]) AND psychomotor agitation[Title] AND children[Title]”; (21) “(lithium[Title]) AND psychomotor agitation[Title] AND adolescents[Title]”; (22) “(valproate[Title]) AND psychomotor agitation[Title] AND children[Title]”; (23) “(valproate[Title]) AND psychomotor agitation[Title] AND adolescents[Title]”; (24) “(carbamazepine[Title]) AND psychomotor agitation[Title] AND children[Title]”; (25) “(carbamazepine[Title]) AND psychomotor agitation[Title] AND adolescents[Title]”; (26) “(antihistaminics[Title]) AND psychomotor agitation[Title] AND children[Title]”; (27) “(antihistaminics[Title]) AND psychomotor agitation[Title] AND adolescents[Title]”.

We found a total of 5300 records. There were 1415 records screened after removing duplicates which were not related; of those, 1141 records were excluded because they were not pertinent to the selected topic. A total of 274 full-text articles were assessed for eligibility. A total of 239 full-text articles were excluded because they were not focused on PA and did not totally meet the selection criteria. Finally, a total of 35 reports were included in our qualitative synthesis (for a detailed description, see Figure 1).

### 2.4. Study Selection

Two authors (MGC and BT) independently screened the resulting articles for their methodology and appropriateness for inclusion. Non-controlled studies as well as studies that did not consider the treatment response as a primary outcome were included, considering the paucity of available findings. Consensus discussion was used to resolve disagreements between reviewers.

### 2.5. Data Collection Process and Data Items

First, the title and abstract of each paper were assessed by two independent authors (MGC and BT) for language suitability and subject matter relevance, and the studies thereby selected were assessed for their appropriateness for inclusion and quality of methods. The first author, year of publication, design, sample age, duration of follow-up, intervention and main findings are summarized in Table 1.

### 2.6. Data Synthesis

Due to the lack of homogeneity among the resulting studies, a meta-analysis could not be performed. In particular, studies varied in terms of how improvements were measured. Hence, this systematic review is presented as a narrative synthesis.

## 3. Results

Forty-two reports (42/5401; 0.78%) were included in the review, as summarized in Table 1. We found: eleven case series (11/42, 26.19%), eight chart reviews (8/42, 19.05%), eight case reports (8/42, 19.05%), six double-blind placebo-controlled randomized studies (6/42, 14.29%), four double-blind controlled randomized studies (4/42, 9.52%), four open-label trials (4/42, 9.52%) and one case control (1/42, 2.38%).

The selected studies included an overall sample of 1924 (604 females; 604/1924, 31.39%). The weighted average age was 10.47 years (range from 7 months to 21 years).

### 3.1. Diagnostic Assessment

A total of 842 patients (842/1924; 43.76%) were observed in an Emergency Room or a Psychiatric or Pediatrician Intensive Care Unit and did not receive a formal Diagnostic and Statistical Manual of Mental Disorders (DSM) or International Classification of Diseases (ICD) diagnostic assessment. A total of 1082 patients (1082/1924; 56.24%) were diagnosed as follows: 204 BD (204/1082; 18.85%); 12 MDD (12/1082; 1.11%); 13 Mood Disorder Not Otherwise Specified (NOS) (13/1082; 1.2%); 323 Psychotic Disorder (323/1082; 29.85%); 1 Obsessive Compulsive Disorder (OCD) (1/1082; 0.09%); 1 PTSD (1/1082; 0.09%); 41 Substance Use Disorder (SUD) (41/1082; 3.79%); 314 ASD (314/1082; 29.02%); 49 ADHD (49/1082; 4.53%); 1 Tourette’s Syndrome (1/1082; 0.09%); 17 Neurodevelopment Disorder NOS (17/1082; 1.57%); 145 Disruptive, Impulse-control and Conduct Disorders (145/1082; 13.40%); 7 Adjustment Disorder (7/1082; 0.65%); 1 Intellectual Disability (1/1082; 0.09%); 39 Epilepsy (39/1082; 3.6%); 1 Anti-NMDAR Encephalitis (1/1082; 0.09%).

### 3.2. Pharmacological Treatment

The psychopharmacological treatments administered were heterogeneous, mainly as a consequence of the large number of different diagnoses related to PA.

#### 3.2.1. Benzodiazepines

One prospective, double-blind, RCT compared the effects of preoperative midazolam and ketamine IV injection on Emergency Agitation (EA) after sevoflurane anesthesia in children. Sixty-seven patients (thirty-six females, age range: 2–6, mean age/SD: 4.18 ± 1.33) undergoing ophthalmic surgery were allocated to receive premedication with either 0.1 mg/kg midazolam or 1 mg/kg ketamine. Premedication with 0.1 mg/kg midazolam or 1 mg/kg ketamine lowered preoperative anxiety. Ketamine was very efficient and more effective than midazolam in the prevention of early post-operative emergency agitation and reduced the requirement for rescue medication in children after sevoflurane anesthesia. The study, however, presented a number of limitations: the use of non-specific tools to analyze and measure pediatric EA, the lack of a placebo group (due to ethical issues) and the difficulties in distinguishing between signs of EA and postoperative pain [30]. Similarly, another prospective RCT studied the effects of combining hydroxyzine and midazolam on sevoflurane-induced emergence agitation in pediatric patients undergoing infra-umbilical surgery with a caudal block. Eighty-four children (1–7 years) were assigned to two groups: group M (n = 42), premedicated with 0.5 mg kg oral midazolam, and group MH (n = 42), premedicated with 0.5 mg kg oral midazolam and 1 mg kg hydroxyzine, given 30 min before anesthesia induction. The incidence of sevoflurane-induced emergency agitation was significantly lower in children premedicated with the midazolam and hydroxyzine combination compared to those premedicated with midazolam only. Furthermore, the midazolam and hydroxyzine combination provided better premedication quality than midazolam alone. Nevertheless, the lack of a group premedicated only with hydroxyzine makes it impossible to know whether the effect is additive or synergistic [32]. A single-center study evaluated the efficacy of oral clonazepam versus oral lorazepam, following initial parental diazepam administration (0.2 mg/kg), in managing methamphetamine-induced agitation in children. The authors showed that, although the treatment with oral clonazepam and oral lorazepam is comparable in terms of efficacy in the resolution of agitation, it would be preferable to use lorazepam, as it is less powerful and therefore potentially safer and more manageable [20]. Intranasal lorazepam (2 mg/mL) was administered to a 7-year-old boy at a dose of 1.5 mg (0.05 mg/kg) using a mucosal atomization device, with the total volume divided in half and administered into both nares. The patient, with a history of anxiety and Oppositional Defiant Disorder (ODD), presented to the emergency department with aggression and violent behaviour at home; at 5 min after intranasal lorazepam, he became calm and more cooperative and he was no longer physically or verbally aggressive. However, approximately 90 min after receiving intranasal lorazepam, the patient began kicking and biting the security guards; thus, intramuscular haloperidol and diphenhydramine were successfully administered at that time [17]. Finally, a case report described a 4-year-old-girl who received oral midazolam and developed a paradoxical reaction, which was reversed successfully with flumazenil. Fifteen minutes after midazolam and prior to preparatory activities for lidocaine injection for a laceration repair, the patient began to flail, kick, scream and writhe. Her heart rate increased to 164 beats/min, and her respiratory rate increased to 34 breaths/min. Blood pressure measurement attempts were unsuccessful, but the room air oxygen saturation remained at 100%. Efforts to calm the patient through verbal reassurance by parents, nurses and physicians were unsuccessful. Flumazenil 0.2 mg was rapidly administered intravenously in the next minute, and the patient immediately began to calm down; over the next 15 min, she had completely returned to baseline behavior and personality. Her heart rate decreased to 105 beats/min, and her respiratory rate decreased to 20 breaths/min. After 1 h, the patient was active, oriented and playful. This case highlights, once again, how the clinic must be cautious for possible paradoxical reactions in the administration of BDZ in the pediatric population, especially when neurodevelopmental disorders are involved [27]. The BDZ used in emergency departments can be useful in preventing anesthetic-induced PA episodes; however, given the well-known adverse reactions, they must be used with care, particularly if they are used on underage patients.

#### 3.2.2. Valproic Acid

One of the most studied drugs for PA in children and adolescents is valproate. According to the data collected, forty-one patients have been treated with valproic acid (41/1432; 2.86%), seven inpatients intravenously (7/41; 17.07%), with a dramatic improvement of PA (according to the CGI-BP aggression subscale score), and 34 by oral intake (34/41; 82.9%). Six patients have been treated in an add-on to second-generation antipsychotics (SGAs)/BDZs. Two patients reported acute dermatological side effects, such as skin rash. The sample of thirty-four patients treated orally was included in a prospective 6-month open trial and reported the following side effects: twenty patients had a significant weight gain (20/34; 58.8%), and six patients showed an increase in alanine-aminotransferase (ALT) (6/34; 16.6%). The mean age of the sample was 12.3 ± 3.7 years; the baseline diagnosis was mixed mania; the mean daily dose was 950 ± 355 mg. This study provided evidence for the effectiveness and safety of valproate in the treatment of pediatric mixed mania over a 6-month period [37]. These results must be interpreted in light of several limitations, such as the lack of a control group and the open trial design: although valproic acid was the predominant drug in the trial, 38% of the subjects were also on stimulants.

A case series of six subjects aged 16–17 years suffering from mania with PA showed the efficacy of the intravenous administration of valproate (dose range: 1200–2000 mg), even if in combination with SGA and BDZ. Nausea, vomiting, easy bruising and tremor were the most frequent side effects. Despite the absence of a control group, these findings suggested that, in emergency psychiatry clinical settings, valproate IV is effective in reducing agitation and aggressive behaviors, with a good tolerability profile and improved treatment compliance [15]. A previously described case report of an 8-year-old autistic child in an acute state of agitation reported the same efficacy in the administration of valproate intravenously at a dose of 2000 mg. Valproate IV was effective for this patient with AD for both acute and chronic behavioral agitation [26]. There is also a case report of a patient who exhibited an improvement in PA by suspending the intravenous administration of valproic acid (1000 mg/day for 2 weeks) and by orally taking quetiapine with increasing doses over 4 weeks, reaching a total daily dose of 600 mg. Paradoxical agitation may be a rare side effect of valproate and should be considered when new onset or worsening agitation occurs during its use [13]. It can therefore be said that this drug has shown efficacy in the treatment of acute PA and in the reduction of the frequency of episodes of PA in patients with chronic agitation, maintaining a good profile of tolerability and efficacy.

#### 3.2.3. Risperidone

One of the most utilized drugs in clinical practice is Risperidone. A post hoc analysis of a 6-week, multicenter, double-blind, randomized, parallel-arm trial comparing 6 weeks of administration of risperidone versus placebo in children with disruptive behavioral disorders (DBD) and subaverage intelligence was performed. One-hundred and ten patients were included (twenty-one females), forty-nine on risperidone and sixty-one on placebo. Reported symptoms were assessed along three dimensions: explosive irritability; agitated, expansive, grandiose mood; depression. Treatment effect analysis found that the mean scores of all three dimensions were significantly reduced with risperidone compared with placebo at weeks 2, 4 and 6. However, the sample was heterogeneous and included children with several DBDs and subaverage intelligence. Moreover, the authors derived the assessment of mood symptoms from an assessment originally conceived for the assessment of disruptive symptoms in children (Nisonger Child Behavior Rating Form) [16].

An observational descriptive study was conducted on a sample of seventeen subjects (7 females, age 12.7 ± 2.7) with a diagnosis of neurodevelopmental disorder recruited in a residential inpatient facility, in which problematic behaviors (aggression, agitation, self-injury) and sleep disturbances were assessed in relation to the decalage in treatment dosage. The treatments included risperidone (n = 8), valproic acid (n = 6), methylphenidate (n = 4), trazodone (n = 3), quetiapine (n = 3), lithium (n = 2), oxcarbazepine (n = 2), ziprasidone, olanzapine, molindone, zomisamide and topiramate (n = 1). Twelve patients experienced only decreases in dosage medication. Ten patients displayed higher rates of problem behavior and sleep disturbances during the weeks that followed a medication decrease compared to during weeks without medication changes. The data suggested that tapering in risperidone and trazodone was more likely to increase aggressive behaviors than tapering in valproic acid. However, to define a relationship between the symptoms’ worsening and the dosage tapering of a specific drug was problematic because the patients were treated with multiple drugs, thus limiting the interpretation of the results of the study [39].

The association of piracetam (a positive modulator of AMPA-sensitive glutamate receptors) with risperidone for the treatment of aggressive behaviors and the management of PA was tested with a 10-week, double-blind, placebo-controlled study in a sample of forty outpatients (thirty males, ten females, Mean Age 6.83 ± 1.81) with a diagnosis of Autistic Disorder and related severe disruptive symptoms. The patients were randomly allocated to risperidone + piracetam (Group A) or to risperidone + placebo (Group B). The main outcome measure was the total score of the Aberrant Behavior Checklist-Community (ABC-C) Rating Scale [53]. A significant difference was observed in the ABC-C score changes at week 10 compared with baseline in the two groups, showing a greater efficacy in the treatment piracetam arm when compared to the placebo. Again, the limitations were the small number of patients enrolled and the short follow-up [11].

Sabuncuoglu et al. (2007) described a case series of three children who developed severe adverse reactions after switching from risperidone to methylphenidate. The first patient was a 6-year-old boy, diagnosed with ADHD and ODD. He developed severe hyperactivity and agitation upon taking methylphenidate after the discontinuation of risperidone treatment. A 6-year-girl of with a diagnosis of ADHD and borderline intellectual functioning displayed severe hyperactivity, agitation and irritability upon switching to methylphenidate medication. The third patient was a 15-year-old female adolescent with a similar clinical course as the previous patients. In all cases, methylphenidate discontinuation improved agitation and irritability [43].

Risperidone has been shown to be relatively safe in the treatment of PA in the underage patient, with the development of side effects that led to a decrease in the dose or to its replacement with another drug. During the switching of risperidone, one must also pay attention to the additional side effects that may occur.

#### 3.2.4. Olanzapine

An open clinical trial of olanzapine was conducted on five hospitalized children (age range: 6–11 years, 9.2 ± 2.0, three females and two males) with BD, psychosis NOS and ADHD. The mean length of treatment with olanzapine was 33 ± 19 days, and the mean daily dose was 7.5 mg/day (range: 2.5 to 10 mg/day). All children experienced adverse effects, namely, sedation (n = 3), weight gain of up to 16 pounds (n = 3) and akathisia (n = 2). Three patients showed some clinical improvement, but olanzapine treatment was discontinued in all five children within the first 6 weeks of treatment, mainly because of adverse effects. Improvement was observed in sleep in all five patients and in the control of aggression in three of them. The study was interesting considering that the sample, even if small, consisted of severely afflicted and hospitalized patients with different diagnoses and various symptom presentations, and they were resistant to previous treatments. Unfortunately, no standardized rating scales were used to assess diagnoses, symptoms and treatment outcomes [33]. Sheikh and Ahmed described the case of a 10-year-old-girl with severe acute agitation and aggressive behaviors treated with olanzapine in a pediatric inpatient setting. No psychotic features were reported by the patient or described by the family. Her admitting diagnosis was ADHD comorbid with ODD. At admission, her medication included sertraline (50 mg/day) and amphetamine/dextroamphetamine (10 mg twice/day), suspended after hospitalization. The patient was treated with several “as-needed” medications, including diphenhydramine (up to 150 mg orally/intramuscularly per day), hydroxyzine (up to 100 mg IM per day), lorazepam (up to 2 mg orally/intramuscularly per day) and haloperidol (up to 8 mg orally/intramuscularly per day), with partial clinical benefits. Finally, she gradually started olanzapine up to 10 mg/day with favorable results in a few days and without experiencing significant adverse events in the following 15 days [46]. An open-label trial with olanzapine was conducted on twenty-three subjects between the ages of 5 and 14 years with a diagnosis of manic episode with aggressive behaviors. The patients were treated with Olanzapine (range: 2.5–20 mg; mean dosage: 9.6 ± 4.3 mg/day) for 8 weeks. Two subjects (2/50; 4.0%) reported transient side effects after the acute administration of olanzapine, namely, itching and pseudo-parkinsonism. A significant improvement in the Young Mania Rating Scale (YMRS) [54] aggression subscale score was found. They described, during the 8-week follow-up, a number of well-known side effects, namely: fourteen patients reported an increased appetite (14/23, 60.9%); ten patients reported somnolence (10/23, 43.5%); seven patients reported abdominal pain (7/23, 30.4%); seven reported weight gain (7/23; 30.4%); six reported depressive symptoms (6/23; 26.1%); five reported diarrhea (5/23; 21.7%); five patients reported infection (5/23, 21.7%); five reported fever (5/23, 21.7%). The small sample size, the design of the study and the concomitant use of other medications, such as lorazepam, guanfacine and clonidine, limited the generalizability of results [22]. The efficacy and the safety of olanzapine were also investigated in a group of two hundred and eighty-five pediatric emergency patients. The primary indications for olanzapine administration included agitation (n = 166), headache (n = 58), nausea/vomiting/abdominal pain (n = 37), unspecified pain (n = 20, 7%) and other reasons (n = 4). The route of olanzapine administration was intramuscular (n = 160; median dose, 10 mg), intravenous (n = 101; median dose, 5 mg) and oral (n = 24; median dose, 10 mg). Within the agitated group, thirty-eight required further sedation, and only twenty-eight received additional sedation within 1 h. It was shown that olanzapine, particularly IV or IM, was effective and safe for acute agitation in this special population [18]. A 17-year-old man with a diagnosis of anti-N-methyl-D-aspartate receptor (NMDAR) autoimmune encephalitis, manifesting an altered mental status such as bizarre behaviour and psychomotor agitation, was successfully treated with intramuscular olanzapine 2 mg, as needed, which was later changed to oral olanzapine 2.5 mg twice a day and five rounds of plasmapheresis (over the course of 10 days), with a complete resolution of his altered mental status [42]. An olanzapine-induced episode of anti-cholinergic agitation and delirium has been described in a 6-year-old male patient with ADHD who unintentionally ingested 15–20 olanzapine (5 mg) tablets. In this case, the discontinuation of olanzapine was followed by the administration of intravenous physostigmine (dose: 19.5 mg/day) to reverse anticholinergic syndrome [23].

In summary, olanzapine seems to have a partially favorable profile, in terms of tolerability and efficacy, when administered for the management of aggression and PA in youths, at least in the short term. Long-term treatment is limited by some annoying side effects for the patient, such as weight gain and sedation.

#### 3.2.5. Ziprasidone

One of the most studied drugs is ziprasidone, utilized in monotherapy for one hundred and forty patients (140/1432; 9.77%) or in an add-on to other antipsychotics, mood stabilizers or benzodiazepines (BDZs) in one hundred and twenty-one patients (121/1432; 8.44%), for a total of two hundred and sixty-one patients (261/1432; 18.22%). More in detail, two hundred and twenty-nine patients on ziprasidone were treated with an intramuscular administration (229/261; 87.73%); thirty-two patients (32/261; 12.26%) were administered with an oral intake. Two hundred and forty-seven patients (247/261; 94.63%) responded to treatment with a significant reduction in PA. The clinical response was inadequate in fourteen patients (14/261; 5.37%); ten patients reported significant side effects (10/261; 3.83%), namely, drowsiness, dizziness, nosebleed, general aches, sore muscles, syncopal episode, nausea, stiffness joints, generalized itching and tardive dyskinesia. The first observation with ziprasidone was a prospective open-label trial conducted by McDougle et al. (2002) in which a preliminary evaluation of the safety and effectiveness of ziprasidone in children, adolescents and young adults with neurodevelopmental disorder was assessed. A sample of twelve patients (two females, mean age ± SD, 11.62 ± 4.38 years; range, 8–20 years) with DSM-IV-defined autism (n = 9) or Pervasive Developmental Disorder (PDD) NOS (n = 3) who received ziprasidone (mean daily dose, 59.23 ± 34.76 mg; range, 20–120 mg) for at least 6 weeks (mean duration, 14.15 ± 8.29 weeks; range, 6–30 weeks) was recruited. Six of the twelve patients were considered responders based on a Clinical Global Impression scale (CGI) [55]. In general, ziprasidone was well tolerated by four patients who experienced no adverse effects. Among the eight other patients, sedation was the most common side effect; in most cases, it was transient in nature. One patient developed an oral dyskinesia. This patient had a history of tardive dyskinesia involving his hands. It is possible that the dyskinesia was secondary to the withdrawal of quetiapine. The dyskinesia resolved upon the discontinuation of ziprasidone. Another patient showed transient ‘lisping’. No cardiovascular side effects, including chest pain, tachycardia, palpitations, dizziness or syncope, were observed or reported. The mean change in body weight for the group was −5.83 ± 12.52 lb (range, −35 to +6 lb). Five patients lost weight, five had no change, one gained weight and one had no follow-up weight obtained beyond the baseline measurement. In this study, Ziprasidone appears to have potential for improving symptoms of aggression, agitation and irritability in children, adolescents and young adults with autism. The findings of this study need to be assessed against a few limits: the small sample size with the inclusion of a wide age and IQ range, the unclear impact of comorbid conditions, the lack of a gold standard diagnostic instrument, the unblinded and uncontrolled nature of the trial, the use of a range of concomitant medications, the highly variable lengths of treatment and the lack of a systematic, prospective methodology including a frequent assessment of change in specific maladaptive signs, symptoms and behaviors associated with autism and other PDDs [35]. In a case series, three patients were treated with ziprasidone IM for aggressive behaviors (10–20 mg/day; age range: 12–13 years) [25]. Patients were diagnosed as having ADHD, oppositional defiant disorder and BD. Ziprasidone was effective in treating PA, but syncope occurred in a patient. This case series reinforced the need for effective “as needed” (pro re nata—PRN) medications, which can be used as de-escalating agents when other psychosocial interventions and oral PRN medications have failed [25].

A retrospective chart review of all child and adolescent inpatients was conducted at an acute care private psychiatric hospital in central New York to identify patients who had received intramuscular ziprasidone alone (without other acute agents, such as lorazepam) during the first 9 months of 2003. Forty-nine injections were administered to forty-nine patients (thirty-two females). Forty-three injections were 20 mg, six injections were 10 mg and re-administration within 4 h was required in only one subject. No adverse events were recorded. This study confirmed the high tolerability and efficacy of ziprasidone for acutely agitated and aggressive inpatient youth, but limitations influenced the study results, namely, the reliance on nursing-shift notes, which may be brief, especially in times of acute agitation in a busy inpatient unit, and the lack of predetermined formal assessment parameters (such as pre-injection medications and interventions, post-injection time to calming, post-injection over sedation and the measurement of vital signs) [47].

Khan & Mican (2006) [29] conducted a chart review on a total of one hundred patients with PA (age range: 12–17 years) treated with ziprasidone IM (20 mg/day) or olanzapine IM (5–10 mg/day). No significant differences emerged between the two treatments, both on safety and efficacy. The results suggested that ziprasidone IM and olanzapine IM might be equally effective in treating aggression in youth. These agents were also similar with regard to safety, because no clinically significant adverse events were reported for either treatment group. Due to the retrospective nature of this study, there were several limitations: the findings depended on existing medical records for data collection; no definitive conclusions on safety in children and adolescents could be made because of the poor documentation of adverse events in this study; there was a lack of standardized efficacy measures; subjects between study groups were not matched with respect to characteristics such as type of psychiatric illness, severity of illness, duration of hospitalization and Axis I comorbidities [29]. A second chart review included fifty-nine patients with several psychiatric diagnoses (BD, major depressive disorder, psychotic disorder, disruptive disorder, impulse control disorder, ADHD; age range: 5–19 years) treated with ziprasidone IM 10–20 mg/day. A total of 81% (n = 48) of the selected cases improved, but with a relevant burden of side effects (such as an increase in seizure frequency, dizziness, nosebleeds, sore muscles, general aches and confusion). Ziprasidone IM was helpful for agitation but caused over-sedation. According to the authors, safety data (firstly, ECGs), should become part of the clinical routine and be assessed in controlled prospective studies. Although this study has a good sample size and standardized ratings, there were several limitations, apart from the lack of ECG data. First, the study design might have biased the evaluating psychiatrists in their assessments of the effectiveness and tolerability of IM ziprasidone. Second, the treatment group had a wide age range (5–19 years) and was mainly constituted by males [14].

In a retrospective case-control chart-review, fifty-two patients (age range: 12–17 years) were treated with ziprasidone IM (dose range: 10–20 mg/day) vs. haloperidol IM (dose range: 2.5–10 mg/day) + lorazepam IM (1–2 mg/day). No differences were found between the two groups. Ziprasidone IM was effective, well tolerated and similar in its clinical profile to more usual medications IM for treating severe agitation in adolescents, such as first-generation antipsychotics in monotherapy or combined with lorazepam. Again, the methodological limitations of this study included potential retrospective biases, a lack of more extensive results from the standardized rating scale of agitation and a modest sample size [28]. A chart-review study on twenty patients (aged 9 months to 17 years) who developed PA following traumatic brain injury showed that ziprasidone IM 0.07–1.7 mg/kg was safe and effective. Ziprasidone was safe and effective in pediatric patients with closed head injuries who developed agitation and/or aggression in the postinjury period. Based on this limited patient series, ziprasidone consistently lowered the Sedation Agitation Scale (SAS) [56] scores in all age groups. There were minimal dose adjustments, the duration of therapy was brief and no adverse events were reported [45]. A more recent chart review on ziprasidone IM (range: 2.5–20 mg/day) administered to forty patients (age range: 5–18 years) found a positive response at an average dosage of 0.19 ± 0.1 mg/kg. Thus, non-responders received a low dosage of 0.13 ± 0.06 mg/kg. Moreover, a significant dose difference was found between patients who required only one initial dose of ziprasidone compared to those who required additional medication. An initial dose of 0.2 mg/kg of ziprasidone IM should be taken into account in acutely agitated pediatric patients. However, a valid scale for the agitation assessment was not applied, patients’ home medications and previous treatments with atypical antipsychotics were not collected (this could have altered the emergency department provider’s dose selection and patient response), the assessment time was short and, finally, no patients received ECGs for the assessment of QT/QTc [36]. Taken as a whole, studies on ziprasidone, even if of limited evidence, demonstrated a good profile of efficacy and safety in the acute condition, with few side effects, in a wide dose range (between 2.5 and 20 mg/day). Long-term data are lacking. ECG monitoring is warranted.

#### 3.2.6. Aripiprazole

One hundred and eighteen children and adolescents (aged 6–17 years) with AD tantrums, aggressions and self-injurious behaviors were randomized to aripiprazole (5, 10 or 15 mg/day) or placebo in an 8-week double-blind, randomized, placebo-controlled, parallel-group study. Efficacy was evaluated using the caregiver-rated Aberrant Behavior Checklist (ABC) Irritability subscale (primary efficacy measure) and the clinician-rated CGI. At week 8, all aripiprazole doses produced a significantly greater improvement than placebo in mean ABC Irritability subscale scores and in mean CGI scores. The discontinuation rates due to adverse events were as follows: placebo 7.7%, aripiprazole 5 mg/day 9.4%, 10 mg/day 13.6% and 15 mg/day 7.4%. The most common adverse event leading to discontinuation was sedation. There were two serious adverse events: presyncope (5 mg/day) and aggression (10 mg/day). Although it is not possible to draw definitive conclusions due to the short-term nature of the study and the lack of a control group with other drugs, it would seem that aripiprazole was effective, generally safe and well tolerated [34].

A post hoc analysis evaluated the effects of aripiprazole on Positive and Negative Syndrome Scale (PANSS) [57]-derived Hostility factor scores in adolescents with schizophrenia. A sample of 302 adolescents (13–17 years) with schizophrenia was enrolled in a 6-week, multicenter, double-blind, randomized, placebo-controlled trial comparing aripiprazole (10 or 30 mg/day) with placebo. After 6 weeks, aripiprazole 10 mg/day and aripiprazole 30 mg/day showed a statistically significant improvement versus placebo. However, the post hoc nature of the analyses limited the results’ strength; moreover, investigators did not control for secondary medication effects such as adverse events (e.g., sedation) or clinical improvements other than hostility control (e.g., changes in positive symptoms) [41].

The use of aripiprazole seems to be effective in reducing the frequency of PA; however, more data are needed. It is also necessary to consider the serious side effects that may occur.

#### 3.2.7. Haloperidol

A case series explored sertraline (25–75 mg for 2–8 weeks) as an add-on to haloperidol and other drugs in nine patients aged 9–12 years with ASD, anxiety and PA. These patients were particularly intolerant to changes in their routines, transition-induced agitation, irritability and panic. Eight patients showed a good clinical response, with only 1 drop due to stomachache, but with possible behavioral worsening at 75 mg/d. This study suggested that low-dose, short-term sertraline treatment with divided doses during the day could lower behavioral reactions when in association with situational transitions or environmental changes in children with autistic disorders, though the beneficial effect might be only temporary [48]. However, in these cases, sertraline should be administered cautiously, with low titration, as SSRIs are not rarely associated with increased irritability and hostility in children with ASD [58,59]. In a subsequent case series, the administration of intravenous haloperidol resulted in prompt behavioral control in five difficult-to-sedate, critically ill children (four ventilated). In addition to the sedative-sparing effect, haloperidol appeared to facilitate ventilator weaning. In four of the five patients, haloperidol was well tolerated, without any short- or long-term side effects. Case 5 was a patient with an oculogyric crisis, a form of dystonic reaction characterized by a sudden involuntary upward deviation of the eyes. The authors suggested that intravenous haloperidol might be useful only in carefully selected, agitated, critically ill children considering its difficult-to-manage tolerability profile [24]. The effectiveness and safety of haloperidol were investigated by Ratcliff et al. in a retrospective chart review of eight hundred and fifty-five acutely ill children, twenty-six of whom received haloperidol, by documenting adverse effects after the drug administration. The main use of haloperidol was for severe agitation and restlessness (85%). Less frequently, haloperidol was administered to treat delirium with disorientation, hallucinations and delusions (15%). Like typical critically ill trauma and acute burn patients, these patients were treated with various combinations of pain and anxiety medications before and/or concurrent with the administration of haloperidol. The largest cumulative dose administered during a 24 h period was 0.957 mg/kg. The longest period of treatment with haloperidol was twenty-two days. Six patients out of twenty-six (23%) showed significant adverse reactions, such as hyperpyrexia, and dystonic reactions [40]. Sixty-nine patients who received at least one pro re nata (PRN) oral dose of immediate-release (IR) quetiapine, haloperidol, loxapine or chlorpromazine for acute agitation and aggression, without regard to the aetiology of symptom presentation, were analyzed retrospectively. The mean haloperidol, loxapine, chlorpromazine and quetiapine doses were 4 mg, 13 mg, 29 mg/dose and 32 mg/dose, respectively, and the response rates were 36%, 30%, 50% and 53%, respectively. Considering the response rate and the absence of induced extrapyramidal effects (EPS), the authors encourage the use of quetiapine in this subpopulation of patients [52]. A sample of 24 adolescent patients treated in an Emergency Room with haloperidol and BDZ showed a reduction in PA/aggressiveness. No significant side effects were reported. As said before, haloperidol + BDZ showed an effective action comparable to ziprasidone IM [28].

Therefore, the use of haloperidol, despite its effectiveness, can lead to the development of important adverse events that make its tolerability profile rather poor.

#### 3.2.8. Other Pharmacological Treatments

A limited number of other drugs have been tested in anecdotal reports.

A retrospective case series was performed to examine the response to oxcarbazepine prescribed for irritability/agitation symptoms in a sample of thirty patients (age range: 5–21, mean age: 12.0 years) with Autism Spectrum Disorder (ASD). Oxcarbazepine was an add-on to atypical antipsychotics, SSRIs, alpha agonists and valproic acid. Eighteen patients continued to take oxcarbazepine at the end of the study period out of the thirty patients who started oxcarbazepine during the retrospective study. For these patients, the average final daily dose was 1360 mg (range 600–1800). Twelve patients stopped oxcarbazepine for adverse events (one for hyponatremia with associated seizures, one for new onset generalized seizures, one for an allergy and four for a worsening of irritability) or for a lack of a perceived benefit. The patients had an overall benefit from the add-on treatment with oxcarbazepine, with a good management of irritability, aggression, agitation and inner tension. However, a high rate of adverse events was found, suggesting caution in the administration of oxcarbazepine in this specific group of patients [19].

As part of clinical trials and an ongoing protocol evaluating the benefit of carbamazepine (CBZ) as a treatment for aggression in inpatient children and adolescents, six boys (aged 10 to 16 years) were found to develop adverse behavioral and neurological reactions. While on CBZ, one adolescent became manic (first report of CBZ-induced mania in an adolescent); one developed hypomania; two showed increased irritability, impulsivity, hyperactivity and aggression; one developed an abnormal EEG with sharp waves and spikes; and one had the first recurrence in years of an absence of seizures. Various mechanisms for these seemingly paradoxical effects have been hypothesized, including a “switch” process similar to tricyclics. [38].

Wakai et al. (1994) described the case of a 4-year-old boy with benign partial epilepsy (BPE) with affective symptoms associated with hyperkinetic behavior, agitation and aggressiveness during interictal periods. The young child presented hyperactivity and restlessness since about age 3. At 4 years, he developed episodes consisting of an expression of terror without a complete loss of consciousness. Although the authors first suspected an acute psychic problem, the ictal EEG was abnormal. After the administration of CBZ, a prompt decrease in the hyperkinetic and aggressive behaviors and a gradual reduction in the frequency of the attacks was noted [50]. In a retrospective chart review, the impact of anticonvulsant therapy in thirty-eight children with bipolar spectrum disorder and epilepsy in comorbidity was analyzed. Two mental health clinicians independently assessed psychiatric diagnoses, symptoms and assigned retrospective CGI ratings for psychiatric symptoms. Thirty patients were diagnosed with complex partial seizure disorder and eight had primary generalized seizure disorder. Nineteen met the DSM-IV criteria for BD-I, and nineteen had Mood Disorder NOS. Forty-two medication trials with 11 different anticonvulsants were identified. Of the thirty instances in which anticonvulsant monotherapy was attempted, CBZ, divalproex sodium, lamotrigine and oxcarbazepine were associated with better psychiatric CGI ratings than other monotherapies. In particular, there was a decrease in intensity in the expression of common bipolar symptoms such as impulsivity, PA and explosive rage. Despite the important methodological limitations due to the nature of the study, the limited sample size and the clinical assessment, the authors suggested that anticonvulsants appeared to simultaneously treat both epilepsy and mood disorder, and they would seem promising in the management of PA, episodes of anger, aggression and impulsiveness [44]. Both CBZ and oxcarbazepine have proven to be sufficiently effective in the control of agitation, impulsivity and aggression, especially if these symptoms are associated with mood disorders. However, they have less efficacy than other anticonvulsants such as valproic acid and a greater risk of adverse events. Four inpatients (4/363; 1.1%) have been treated with methotrimeprazine as an add-on to several other treatments (namely, haloperidol, ketamine, BDZs, alimemazine, apomorphine and chloral hydrate), with no significant side effects and a good response of PA. Patients were diagnosed as having ‘*refractory agitation*’ (aged from 7 months to 15 years) and were administered with a dose range between 3 and 12 mg. Once refractory agitation still continues after adequate 24 h dosages of D2-blocking agents, the addition of methotrimeprazine might be a useful option [49]. Ten inpatients with ADHD (10/363; 2.7%) have been treated with D-methylphenidate. They showed a marked reduction in PA, with no side effects. The results supported the hypothesis that ADHD and Conduct Disorder could in some way share a common pathophysiological pathway: since both hyperactivity and aggression respond to treatment with d-amphetamine, there may be etiological or other factors common to both clinical conditions [12]. Other findings suggest the use of stimulants when aggression is associated with ADHD as a first choice or add-on treatment [60]. A multicenter, randomized, double-blind, placebo-controlled study analyzed the efficacy of lithium monotherapy in the acute management of pediatric or adolescent patients with BD-I/manic or mixed episodes. Pediatric participants (7–17 years) with BD-I/manic or mixed episodes compared lithium (n = 53) versus placebo (n = 28) for up to 8 weeks. There were statistically significant differences in both the YMRS score and the over CGI scores in the group being treated with Lithium compared with that being treated with the placebo. The effectiveness of Lithium in reducing manic symptoms, particularly those of the psychomotor sphere, became apparent after a very short time, with a great tolerance and no side effects recorded [21]. Nineteen patients (aged 10–18 years old) presenting for antimuscarinic toxidrome were randomly treated with physostigmine 0.02 mg/kg bolus followed by a 4 h physostigmine infusion (0.02 mg/kg/h) and lorazepam bolus (0.05 mg/kg) followed by a 4 h normal saline infusion (9 vs. 10 subjects, respectively). There was a significant decrease in agitation scores in the physostigmine arm compared to the lorazepam arm after the initial bolus, but there was no difference at the fourth hour of infusion [51]. The effects of dexmedetomidine on post-operative agitation in pediatric patients who underwent adenotonsillectomy surgery with sevoflurane anaesthesia were found to be greater than those in patients treated with tramadol. The post-operative Riker Sedation-Agitation Scale (SAS) was significantly lower for patients who received dexmedetomidine than it was for patients who received tramadol at the post-operative time points of 1, 15 and 30 min [31].

Other drugs may be useful in the treatment of acute PA, especially if they are in association with certain pathologies, such as methylphenidate in patients with ADHD; however, more data are needed, given the lack of studies present.

## 4. Discussion

The analysis of the 42 selected papers revealed four main findings: (1) Ziprasidone, risperidone and aripiprazole are the most studied drugs for the treatment of PA in children and adolescents; (2) Ziprasidone, risperidone, aripiprazole, olanzapine and valproate, with different timings and methods, have all shown efficacy in the acute treatment of PA, but with limited degrees of evidence; (3) the majority of observations are chart reviews, retrospective case-control studies or anecdotal reports; (4) there is no information on the long-term follow-up of patients acutely treated for PA, except for one study with olanzapine and one study with valproate.

### 4.1. Summary of Evidence

According to the *Center for Evidence-Based Medicine Levels of Evidence for Therapeutic Studies*, psychopharmacological treatments of PA in children and adolescents range from 3B (individual case control studies) to level 4 (case series and poor-quality cohort and case control studies) or 5 (expert opinion) (http://www.cebm.net, accessed on 16 September 2021). There is a preference among clinicians for the use of antipsychotics such as ziprasidone, olanzapine or haloperidol (in monotherapy or in association with BDZ) through the intramuscular route, which show efficacy, a rapidity of action and a fair safety of use. However, orally administered SGAs have been shown to be at least similarly effective and with a higher tolerability and acceptance, limiting the IM route to un-cooperative, non-compliant patients [61,62]. Valproate has been tested mainly in bipolar patients with PA. In terms of efficacy, it is comparable to antipsychotics, but with a safer tolerability profile.

### 4.2. Summary of Limitations

There are several limitations to consider when interpreting this review. The first is that only thirty-six reports met the requirements to be included in our systematic review. Moreover, no study addressed the question of a mid/long-term standardized follow-up, with a maximum observational time of 8 weeks in some studies. The number of diagnoses was high, with inhomogeneous samples, given that agitation is an acute condition crossing multiple diagnostic areas. The mode of action, with its sedative profile, has also not been further investigated. Finally, the temporal evolution of the use of the drugs considered was not taken into account in our synthesis.

## 5. Conclusions and Areas for Future Research

A limited number of reports have addressed the problematic issue of the psychopharmacological management of PA in children/adolescents. Taken as a whole, the evidence is unsatisfactory, even if some antipsychotics (such as ziprasidone, olanzapine or haloperidol) might be useful in PA, as well as valproate, but mainly in young bipolar patients.

The findings of our review identified shortcomings in the study design, including treatment-specific biases arising from small sample sizes, a low grade of evidence in studies’ protocols and a clinical rather than statistical interpretation of results. These limitations derive from the difficulties in conducting research studies with children and adolescents, given their diagnostic heterogeneity, their ethical issues and the challenges in the organization of a mid-term/long-term follow-up.

We believe that, above all, therapeutic choices in emergency settings should be guided by the assessment of the risk/benefit ratio. Due to the lack of data in the literature, further studies will be needed to evaluate the safety profile of medications. Regardless of the pharmacological category used (Antipsychotics, Antihistaminic, BDZ, Mood stabilizers), clinicians should always be aware of the risks related to the dosage required.

## Figures and Tables

**Figure 1 life-13-00293-f001:**
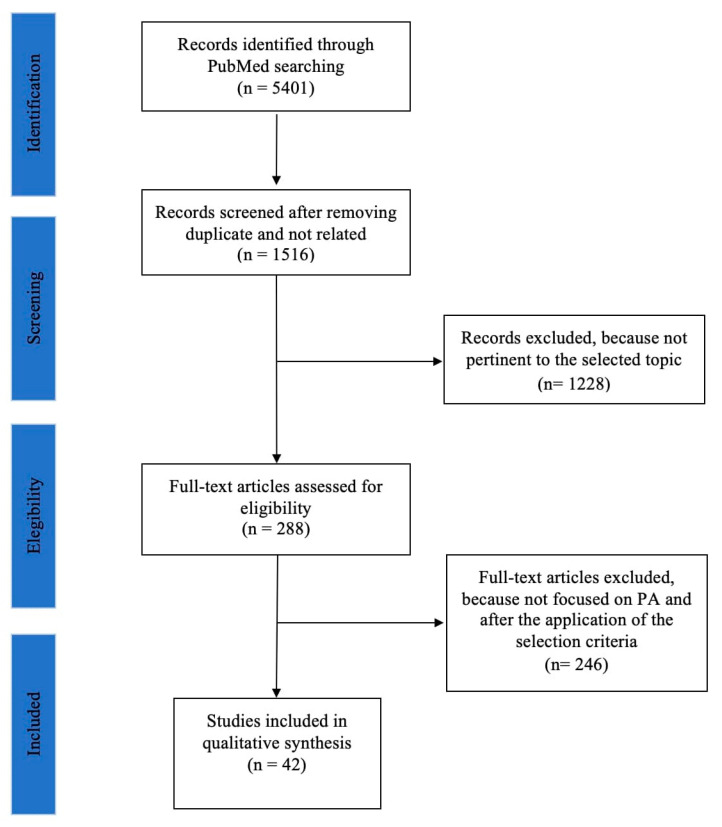
PRISMA flow diagram of selection studies.

**Table 1 life-13-00293-t001:** Summary of selected studies (n = 42).

Author/Publication Year	Study Design	Sample Characteristics	Diagnosis	Treatment	Outcomes
Akhondzadeh et al., 2008 [11]	Double-blind, randomized, placebo-controlled trial	40 pts (10 females) Age range: 3–11 Mean age/SD: 6.83 ± 1.81	ASD	Risperidone vs. Risperidone + Piracetam	The ABC-C Rating Scale scores improved
Amery et al., 1984 [12]	Double-blind, randomized, placebo-controlled trial	10 male pts Age range: 8–11 years Mean age/SD: 9.6 ± 1.6	ADHD	d-Amphetamine	↓ PA
Avari et al., 2016 [13]	Case report	1 male pt (17 years old)	Mood Disorder	Quetiapine administration after valproate suspension	Quetiapine improved disorganized thought and paranoia
Barzman et al., 2007 [14]	Chart review	59 pts (20 females) Age range: 5–19 years Mean age: 13.5 years	Any mental illness	Ziprasidone	↓ PA Side effects reported
Battaglia et al., 2018 [15]	Case series	6 pts (1 female) Age range: 16–17 years Mean age/SD: 16.6 ± 0.4	Mood Disorders Conduct Disorder Substance Abuse	VPA in add-on to SGAs or BDZs	↓ PA Side effects reported
Biederman et al., 2006 [16]	Double-blind, randomized, placebo-controlled trial	110 pts (21 females) Mean age/SD: 8.32 ± 2.3	DBD	Risperidone	Effective in treating the factors of explosive irritability and the management of agitation
Bregstein et al., 2019 [17]	Case report	1 male pt (7 years old)	Oppositional Disorder and Anxiety Disorders	Lorazepam	Transiently effective in treating severe agitation (~90 min)
Cole et al., 2020 [18]	Chart review	285 pts (20 females) Age range: 9–18 years Mean age: 16.4 years	Subjects accessed in the emergency pediatric level I trauma center	Olanzapine	Effective for acute agitation No patient died or had dysrhythmia; one patient experienced dystonia
Douglas et al., 2013 [19]	Case series	30 pts (4 females) Age range: 5–21 Mean age/SD: 12.00 ± 3.6	ASD	Oxcarbazepine	14 significantly reduced the CGI score, 10 reduced the CGI score, 7 discontinued for AE
Farnaghi et al., 2020 [20]	Single-center clinical trial	30 pts (11 females) Age range: 6 months–12 years Mean age: 16 months	Methamphetamine-poisoned children	Clonazepam vs. lorazepam	Clonazepam and lorazepam treatments were equally effective at similar doses. However, considering the higher potency of clonazepam, it seems that lorazepam is the safer treatment.
Findling et al., 2015 [21]	Double-blind, randomized, placebo-controlled trial	81 pts (44 females) Age range: 7–17 Mean age/SD: 11.4 ± 2.9	Bipolar I Disorder	Lithium	↓ in YMRS score and ↓ in CGI score. Lithium was superior to placebo in reducing manic symptoms and the frequency of psychomotor agitation episodes
Frazier et al., 2001 [22]	Open-label trial	23 pts (10 females) Age range: 5–14 years Mean age/SD: 10.3 ± 6.2	Bipolar I-II Disorders	Olanzapine	↓ YMRS aggression subscale scores Side effects reported
Hail et al., 2013 [23]	Case report	1 male pt (6 years old)	ADHD	Physostigmine	↓ agitation and delirium
Harrison et al., 2002 [24]	Case series	5 pts (1 females) Age range: 9 m–16 y Mean age/SD: 8.83 ± 7.35	Critical ill patients (ARDS, peritonitis, graft vs. host, intussusception)	Haloperidol	Reduction in agitation and improvement in ventilator weaning. One patient had a dystonic reaction
Hazaray et al., 2004 [25]	Case series	3 male pts Age range: 12–13 years Mean age/SD: 12.3 ± 0.47	Conduct Disorder; ADHD; Oppositional Disorder; Bipolar I Disorder	Ziprasidone	↓ aggression; syncope reported
Hilty et al., 1998 [26]	Case report	1 male pt (8 years old)	ASD	Valproate	↓ PA
Jackson et al., 2015 [27]	Case report	1 female pt (4 years old)	Laceration in upper lip	Flumazenil to reverse the midazolam paradoxical effect	Paradoxical reaction with psychomotor agitation successfully treated with flumazenil
Jangro et al., 2009 [28]	Case control	52 pts (25 females) Age range: 12–17 years Mean age/SD: 15.7 ± 1.35	Psychotic Disorders; SUD; Adjustement Disorder; Impulse-control Disoder	Ziprasidone, haloperidol, BDZs	↓ PA
Khan and Mican, 2006 [29]	Chart review	100 pts (50 females) Age range: 12–17 years Mean age/SD: 14.5 ± 2.25	Any mental illness	Ziprasidone vs. Olanzapine	↓ PA
Kim et al., 2016 [30]	Double-blind, randomized trial	67 pts (36 females) Age range: 2–6 Mean age/SD: 4.18 ± 1.33	Pediatric patients undergoing ophthalmic surgery	Midazolam and Ketamine following sevoflurane anesthesia	Ketamine was more efficient in ↓ the incidence of emergence agitation at 10 and 20 min after the transfer to the post-anesthetic care unit than midazolam
Koceroglu et al., 2020 [31]	Randomized trial	60 pts (32 females) Age range: 2–9 Mean age: 5.8	Pediatric patients undergoing an adenotonsillectomy using sevoflurane	Dexmedetomidine vs. tramadol	Dexmedetomidine was more effective than tramadol for mitigating post-operative agitation
Koner et al., 2011 [32]	Double-blind, randomized trial	84 pts (12 females) Age range: 1–7 Mean age: 2.4	Pediatric patients undergoing infraumbilical surgery with a caudal block	Midazolam and hydroxyzine	The incidence of sevoflurane-induced emergence agitation was significantly lower in children premedicated with a midazolam and hydroxyzine combination compared to those premedicated with midazolam only
Krishnamoorthy and King, 1998 [33]	Case series	5 pts (3 females) Age range: 6–11 Mean age/SD: 9.2 ± 2.0	Bipolar disorder, ADHD, Impulse Disorder Psychotic Disorder NOS	Olanzapine	↓ aggressive behavior in 3 pts
Marcus et al., 2009 [34]	Double-blind, randomized, placebo-controlled trial	218 pts (23 females) Age range: 6–17 Mean age: 9.7	ASD	Aripiprazole	Improvement in mean Aberrant Behavior Checklist Irritability subscale scores and CGI score
McDougle et al., 2002 [35]	Open-label trial	12 pts (2 females) Age Range: 8–20 Mean age/SD: 11.62 ± 4.38	ASD	Ziprasidone	↓ agitation, irritability and aggressiveness (CGI score)
Nguyen et al., 2018 [36]	Chart review	40 pts (8 females) Age range: 5–18 years Mean age: 11.8	ADHD, ASD, Bipolar I Disorder, PTSD	Ziprasidone	↓ PA 13 non-responders (32%)
Pavuluri et al., 2005 [37]	Open-label trial	34 pts (13 females) Mean age/SD/12: 12.3 ± 3.7	Mixed Episode (Bipolar I Disorder)	Valproate	↓ CGI-BP aggression subscale scores
Pleak et al., 1988 [38]	Case series	6 male pts Age range: 6–10 Mean age/SD: 12.83 ± 2.48	ADHD, Conduct Disorder, Intermittent Explosive Disorder	Carbamazepine	↑ irritability, aggressiveness, impulsivity, manic and hypomanic symptoms, risk seizures
Rapp et al., 2007 [39]	Case series	17 pts (7 females), Age range: 9–17 Mean age/SD: 12.7 ± 2.7	Neurodevelopmental Disorders	Antipsychotics, Antihistaminics, Mood stabilizers	A reduction in the doses of the medications increased aggression, agitation, self-injuries, anger, etc.
Ratcliff et al., 2004 [40]	Chart review	26 pts (7 females) Mean age/SD: 11.7 ± 3.9	Agitated, Acutely Ill Pediatric Burn Patient	Haloperidol	↓ agitation, but 23% had AE (hyperpyrexia or dystonic reaction)
Robb et al., 2010 [41]	Double-blind, randomized, placebo-controlled trial	302 pts (131 females) Age range: 13–17 Mean age: 15.5	Schizophrenia	Aripiprazole	↓ PANSS Hostility, Uncooperativeness and Poor Impulse Control items
Roberts et al., 2020 [42]	Case report	1 male pt (17 years old)	Anti-NMDAR encephalitis	Olanzapine	↓ agitation and irritability and ↑ cooperativity
Sabuncuoglu, 2008 [43]	Case series	3 pts (2 females) Age range: 6–15 Mean age/SD: 9 ± 5.2	ADHD	Switch from Risperidone to Metylphenidate	Increase in PA and aggressive behavior
Salpekar et al., 2006 [44]	Chart review	38 pts (17 females) Age range: 6–17 Mean age: 10.4	Complex partial seizure, primary generalized seizure disorder, Bipolar Disorder	Anticonvulsivants	↑ CGI ratings
Scott et al., 2009 [45]	Case series	20 pts (8 females) Age range: 9 months–17 years Mean age: 7.19	Traumatic brain injury	Ziprasidone	↓ PA
Sheikh and Ahmed., 2002 [46]	Case report	1 female pt (10 years old)	ADHD and ODD	Olanzapine	↓ agitation, irritability and aggressiveness
Staller et al., 2004 [47]	Chart review	49 pts (32 females) Mean age: 17.49	Psychomotor agitation, Agitation/anxiety/threat, Psychotic Disorder NOS	Ziprasidone	↓ agitation and aggressiveness
Steingart et al., 1997 [48]	Case series	9 male pts Age range: 6–12 years Mean age: 8.56	ASD	Sertraline as add-on to haloperidol	8 patients reported ↓ PA; 1 drop-out (stomachache)
van der Zwaan et al., 2012 [49]	Case series	4 pts (1 female) Age range: 7 months–15 years Mean age/SD: 8.4 ± 5.8	Any mental illness	Methotrimeprazine	↓ PA
Wakai et al., 1994 [50]	Case report	1 male pt (4 years old)	Benign partial epilepsy	Carbamazepine	↓ frequency of the attacks and the hyperkinetic behavior
Wang et al., 2021 [51]	Blinded, randomized clinical trial	19 pts (12 females) Age range: 10–18 Mean age: 13.9	Antimuscarinic toxidrome	Physostigmine vs. lorazepam	Physostigmine was superior to lorazepam in controlling antimuscarinic delirium and agitation after bolus dosing
Yip et al., 2020 [52]	Chart review	69 pts (39 females) Age range: 5–16	Agitated or aggressive patients arrived at the emergency department	Quetiapine, haloperidol, loxapine, chlorpromazine	Drugs have comparable efficacy in managing agitation, but quetiapine has a lower risk of inducing EPS

↓ = decrease; ↑ = increase. ABC-C: Aberrant Behavior Checklist-Community; ADHD: Attention Deficit and Hyperactivity Disorder; AE: Adverse Effect; ARDS: Acute Respiratory Distress Syndrome; ASD: Autism Spectrum Disorder; BDZ: Benzodiazepine; CGI: CLinical Global Impression; DBD: Disruptive Behavioral Disorder; EPS: Extrapyramidal Symptoms; NMDAR: N-Methyl-D-Aspartate Receptor; NOS: Not Otherwise Specified; ODD: Oppositional Defiant Disorder; PA: Psychomotor Agitation; PANSS: Positive and Negative Syndrome Scale; PTSD: Post-Traumatic Stress Disorder; SD: Standard Deviation; SUD: Substance Use Disorder; VPA: Valproic Acid; YMRS: Young Mania Rating Scale.

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
