# Peer review of "A Critical Review of the Psychomotor Agitation Treatment in Youth"

_life, 2023, doi:10.3390/life13020293_

Round 1

Reviewer 1 Report

The authors present a review about the treatment option for psychomotor agitation. Despite this being a very important clinical theme the literature and research is relatively limited in this field. This unified view of the current knowledge/clinical practice in the field is of great interest.

The methods used are appropriate and clearly presented. Overall the scientific/clinical interest is high.

However, the authors should revise the text substantially in term of language use. Main problems/inconsistences are found below:

Line 83: probably "findings"?

As a general rule, numbers should be given as numerals (not words) if below 10 (unless they are at the beginning of a sentence). Please revise throughout the text

Lines 210: probably "Emergency Agitation"?

Line 211: authors should specific which rescue medication they refer to

Lines 234 and 253: ED or emergency department -> please harmonize

Line 261: specify the meaning of SGAs

intravenous/intramuscolar is sometimes reported as  IV/IM or i.v./i.m. : please harmonize

Table 1 should contain a list of the abbreviation used and their meaning

Line: 443-444 specific that with p.r.n. authors refer to pro re nata

Lines 670-677 are replicated at 681-687

Line 703: here authors write they did a systematic review, but at the beginning they mention that due to the types of studies included they could only run a narrative synthesis. Please harmonize

Author Response

Reviewer #1: The authors present a review about the treatment option for psychomotor agitation. Despite this being a very important clinical theme the literature and research is relatively limited in this field. This unified view of the current knowledge/clinical practice in the field is of great interest. The methods used are appropriate and clearly presented. Overall the scientific/clinical interest is high.

We thank the reviewer for appreciating our work. This review is intended to be a collection point for information and clinical practices regarding agitation in children and adolescents in view of the growing interest in this topic in the period we are living.

However, the authors should revise the text substantially in term of language use. Main problems/inconsistences are found below:

  1. Line 83: probably "findings"?
    1. We have corrected the word as indicated.
  2. As a general rule, numbers should be given as numerals (not words) if below 10 (unless they are at the beginning of a sentence). Please revise throughout the text
    1. We revised the whole text and corrected following the recommended rule.
  3. Lines 210: probably "Emergency Agitation"?
    1. We have corrected the word as indicated.
  4. Line 211: authors should specific which rescue medication they refer to
    1. These rescue medications refer to other practices used in emergencies to counter psychomotor agitation. Due to the lack of this information in the cited article, it was not possible to be more specific.
  5. Lines 234 and 253: ED or emergency department -> please harmonize
    1. We chose the full form and eliminated the abbreviation.
  6. Line 261: specify the meaning of SGAs
    1. We have correctly introduced the abbreviation SGAs related to second-generation antipsychotics in the Valproic Acid section and correctly used the abbreviation in the remaining text.
  7. intravenous/intramuscolar is sometimes reported as IV/IM or i.v./i.m.: please harmonize
    1. We chose and applied the IV and IM form for the entire text.
  8. Table 1 should contain a list of the abbreviation used and their meaning
    1. We have included a legend at the end of the table of all abbreviations used.
  9. Line: 443-444 specific that with p.r.n. authors refer to pro re nata
    1. We have correctly introduced the abbreviation PRN related to pro re nata in the Ziprasidone section and correctly used the abbreviation in the remaining text
  10. Lines 670-677 are replicated at 681-687
    1. We removed the repetition.
  11. Line 703: here authors write they did a systematic review, but at the beginning they mention that due to the types of studies included they could only run a narrative synthesis. Please harmonize
    1. This systematic review was set up in a narrative style of the data available to us. Due to the heterogeneity of the data, meta-analysis was not possible.

Reviewer 2 Report

A critical review of psychomotor agitation treatment in youth

This paper provides a critical review of pharmacological treatments for psychomotor agitation (PA) in children and adolescents. While it appears to have been initially planned as a meta-analysis the paucity of studies and heterogeneity among studies makes a critical review more warranted at this time. Overall, this review is valuable and largely well done though the addition of the considerations discussed below might increase the value of this project.

The comments below are broken down into those relevant to the scientific merit of the study and those related to the written presentation of this research.

Scientific merit comments:

·       The studies collected for this review were published between 1984 and 2022 and discussed with little attention to issues related to their publication across time. At minimum this limitation should be acknowledged if further analysis is not possible. However, further consideration of temporal issues might be insightful. For example, most atypical antipsychotics were not available until the mid-1990’s and by examining the trends in use of various drugs across studies an understanding of which drugs might have emerged as more frequently administered or effective over time might be gaged.  Have atypical antipsychotics (or a specific drug(s)) supplanted the use of benzodiazepines more recently for PA? A Chi square analysis of studies conducted examining the use of specific drugs or drug classes across time might be informative, if possible.

·       Like the points made above, it would be interesting to know the trajectory of publications across time on this subject. Is there an overall increase/decrease/no change in publications on this topic and what is the trend in use across different treatments over time?

·       This is not an issue of major concern, but the authors used a single data base (PubMed) to search for relevant studies, some research could have been missed by use of a single data base, perhaps other search engines could have been used as well.

·       The authors distinguish use of various medications for acute versus chronic treatment of PA and note the lack of research addressing the latter.  However, the authors should highlight that many of these drugs are highly sedating (especially upon initial use and prior to the development of tolerance) and therefore their acute effects for PA might result from this sedation (or other side effects) rather than direct effects on PA.

·       In the discussion the authors could address general findings regarding treatment of PA in adult populations and whether the patterns they observed in youth are similar.

·       It would be interesting to see if PA is treated differently among those who were observed in Emergency Rooms and Intensive Care Units versus other settings. Similarly comparisons of treatment among those who had acute PA in acute situations (such as surgical reactions) versus those whose PA was associated with a more chronic condition (BD, PD,…) would be of interest to readers.

·       In the Pharmacological treatment section, the authors may want to reconsider how they label/organize the different treatments. The benzodiazepines subsection is fine, a subsection for antipsychotics might be beneficial. The vast majority the other medications highlighted could be characterized as anticonvulsant/mood stabilizers and thus this additional subsection could be used for Valproic Acid, Carbamazepine, etc.

·       The Summary of limitations section could be improved. There are several limitations that the authors mention earlier in the paper that could be revisited here such as the lack of patients with established DSM or IDC diagnoses. Moreover, several of the questions raised above, if not further assessed, should be recognized as limitations such as the lack of information regarding temporal patterns from study publication, the lack of insight provided regarding mechanistic effects – particularly as related to the role of sedation.

·       While overall the study was interesting and informative and I appreciate the challenges the authors had in drawing conclusions from such a variable literature I was left at the end believing the authors could have pulled more conclusions from their study, see above suggestions, and would like to see the authors further consider the results of their critical review in how to move forward move forward in understanding the value of various medications in treatment PA in youth.

Writing suggestions:

The writing quality of this study is generally good. The author’s text was readily understandable, however, some of the writing could be improved. The comments below do not represent a comprehensive list of writing errors, but several things did stand out and I have listed them below.

·       This manuscript is largely written from a first-person perspective (“we” and “our”). This is often a personal choice or an editorial decision, but I ask the authors to consider changing to 3rd person as it has a more professional tone.

·       There seems to be a significant number of very long paragraphs in this paper. Particularly in the Pharmacological treatment section. This makes the material challenging to read and digest, especially when there are multiple studies combined into a single long paragraph. The authors should revisit some of these long paragraphs and break them up to improve readability and clarity of information.

·       Lines (L) 26-28 – list of research methods used in this study should be plural – only the first one in the list is currently plural.

·       L 28 – period should come after the reference

·       L 29 – should “are” be “were”?

·       L 46-49 – should all disorders in this sentence be capitalized?

·       L 49 – should “is” be “has” (past tense- study completed)?

·       L 50 should “outpatients’” be “outpatient”?

·       L 55 “leaded” should be “lead”.

·       L 62 – sentence starting “The majority…” is awkward.

·       L 76 – “is” should be “was”

·       L 93 – the word “initially” in the sentence is a bit confusing as it suggests other methodology was later used and this was not the case.

·       L 101 – “is” should be “was”? Does this sentence require any further explanation?

·       L 139 – should read “because they were not pertinent..”

·       L 141 – should read “because they were not focused...”

·       L 157 – should read “findings”

·       L 163 – should read “methods.”

·       Table 1. – the authors should consider whether the inclusion a key of abbreviated terms is necessary.

·       L 177 – sentence starting “We found… “ the types of methods should be plural – e.g. chart reviews

·       L 200 – the authors use the term inhomogeneous but the term heterogeneous might be preferable as it is more common term

·       L 220 – Should “emergence agitation” be “emergent”?

·       L 253 – “involved” might be a better word than “associated”.

·       L 257-258- Start of the sentence “According to our studies selection” – is awkward – please reword.

·       L 274 – “vomit” should read “vomiting”

·       L 277 – I am a little confused by the use of the phrase “ good compliance” in this context as the treatment was i.v., in an emergency clinical setting, with minors and thus it does not seem like at setting in which compliance would be an issue.

·       L 315 – this sentence might read better “ .. a specific drug was problematic because patients were treated with multiple drugs, thus limiting the interpretation of the results of the study.”

·       L 324 – “scores” should read “score”

·       L 337 – “relatively” may be better than “partially”

·       L 360 – “medication” should read “medications”

·       L 393 – “6- years” should read “6-year”

·       L 415-416 – The beginning of the sentence “It was recruited a sample” is awkward

·       L 426 – add a period after “lisping.”

·       L 439 – should read “were treated with”

·       L 442 – should read “reinforced the need”

·       L 449 – write out a number when it begins a sentence

·       L 457 – “[46]” is the reference number necessary here?

·       L 463 – “finding” should read “findings”

·       L 465 – “prevented to make” is awkward

·       L 512 – Should read “One hundred..”

·       L 537 and L 666 – As noted above the authors often write from a first-person perspective, but here they also use first person (we) to refer to the scientific field which is somewhat awkward

·       L 559 – I am not sure about the necessity of the word “consecutive” here

·       L 607-609 – Across these lines a 4-year old boy subsequently described as an infant – the term “young child” would be more accurate

·       L 628 – “shots” might be replaced with “episodes”

·       L 448 – “BPI”? Do the authors mean BD-I?

·       L 652- delete word “already”?

·       L 659 – spelling of pediatric

·       L 671 – “drug” should be “drugs”

·       L 674 – replace “grade” with “degree”?

·       The Discussion section and the beginning of the Summary of evidence section are identical, this repetition seems unnecessary.

·       L 713 – “AP” should be “PA”

·       L 718 – delete “the” before ethical issues

·       L 722  - term “this aspect” should be clarified

Author Response

Reviewer #2: This paper provides a critical review of pharmacological treatments for psychomotor agitation (PA) in children and adolescents. While it appears to have been initially planned as a meta-analysis the paucity of studies and heterogeneity among studies makes a critical review more warranted at this time. Overall, this review is valuable and largely well done though the addition of the considerations discussed below might increase the value of this project.

We would like to thank the reviewer for considering our paper and its contents well done. As correctly expressed, it was not possible to perform a meta-analysis among the obtained data. However, we believe that the article may be of interest and usefulness to all clinicians.

The comments below are broken down into those relevant to the scientific merit of the study and those related to the written presentation of this research.

Scientific merit comments:

  1. The studies collected for this review were published between 1984 and 2022 and discussed with little attention to issues related to their publication across time. At minimum this limitation should be acknowledged if further analysis is not possible. However, further consideration of temporal issues might be insightful. For example, most atypical antipsychotics were not available until the mid-1990’s and by examining the trends in use of various drugs across studies an understanding of which drugs might have emerged as more frequently administered or effective over time might be gaged. Have atypical antipsychotics (or a specific drug(s)) supplanted the use of benzodiazepines more recently for PA? A Chi square analysis of studies conducted examining the use of specific drugs or drug classes across time might be informative, if possible.

Like the points made above, it would be interesting to know the trajectory of publications across time on this subject. Is there an overall increase/decrease/no change in publications on this topic and what is the trend in use across different treatments over time?

  1. We thank you for the advice, which was undoubtedly constructive and interesting. This article is not intended to demonstrate the temporal evolution of medication use in psychomotor agitation. In our opinion, such analyses would risk changing the meaning of what we would like to communicate. Considering what has been written, the recommended limitation has been added in the appropriate section.
  1. This is not an issue of major concern, but the authors used a single data base (PubMed) to search for relevant studies, some research could have been missed by use of a single data base, perhaps other search engines could have been used as well.
    1. We considered the use of a single database such as PubMed to be sufficiently representative of the studies present in the topic.
  2. The authors distinguish use of various medications for acute versus chronic treatment of PA and note the lack of research addressing the latter. However, the authors should highlight that many of these drugs are highly sedating (especially upon initial use and prior to the development of tolerance) and therefore their acute effects for PA might result from this sedation (or other side effects) rather than direct effects on PA.
    1. Thank you for the suggestion. We believe that not all of the drugs examined are highly sedative and that this sedation is dose related. The purpose of the study is to summarize common clinical data without emphasizing the hypotheses of the mechanism of action of the drugs.
  3. In the discussion the authors could address general findings regarding treatment of PA in adult populations and whether the patterns they observed in youth are similar.
    1. We invite clinicians to the respective local guidelines regarding psychomotor agitation in adults. These are usually not exhaustive regarding the same symptom in young people. This article is intended to be a useful supplement.
  4. It would be interesting to see if PA is treated differently among those who were observed in Emergency Rooms and Intensive Care Units versus other settings. Similarly comparisons of treatment among those who had acute PA in acute situations (such as surgical reactions) versus those whose PA was associated with a more chronic condition (BD, PD,…) would be of interest to readers.
    1. To the best of our knowledge and from what we have learned by conducting this study, the treatment of psychomotor agitation in different settings is similar. Further investigation cannot be performed due to lack of information in the articles reviewed (such as latency of onset or clinical expression).
  5. In the Pharmacological treatment section, the authors may want to reconsider how they label/organize the different treatments. The benzodiazepines subsection is fine, a subsection for antipsychotics might be beneficial. The vast majority the other medications highlighted could be characterized as anticonvulsant/mood stabilizers and thus this additional subsection could be used for Valproic Acid, Carbamazepine, etc.
    1. Thanks for the suggestion. This is a subdivision based on pharmacological class. We chose a subdivision based on individual molecules for easier access to information. For example, studies conducted on Carbamazepine or Oxcarbazepine are few and are not typically used drugs compared to Valproic Acid. Regarding benzodiazepines, we have chosen to group them together because of their similar mode of action.
  6. The Summary of limitations section could be improved. There are several limitations that the authors mention earlier in the paper that could be revisited here such as the lack of patients with established DSM or IDC diagnoses. Moreover, several of the questions raised above, if not further assessed, should be recognized as limitations such as the lack of information regarding temporal patterns from study publication, the lack of insight provided regarding mechanistic effects – particularly as related to the role of sedation.
    1. We have proceeded to add the related limitations.
  7. While overall the study was interesting and informative and I appreciate the challenges the authors had in drawing conclusions from such a variable literature I was left at the end believing the authors could have pulled more conclusions from their study, see above suggestions, and would like to see the authors further consider the results of their critical review in how to move forward move forward in understanding the value of various medications in treatment PA in youth.
    1. We thank the reviewer for the suggested comments. The changes made contribute to give more value to the study performed.

Writing suggestions:

The writing quality of this study is generally good. The author’s text was readily understandable, however, some of the writing could be improved. The comments below do not represent a comprehensive list of writing errors, but several things did stand out and I have listed them below.

  1. This manuscript is largely written from a first-person perspective (“we” and “our”). This is often a personal choice or an editorial decision, but I ask the authors to consider changing to 3rd person as it has a more professional tone.
    1. Since this data is not absolute and unequivocal, on an ever-changing topic, we have tried to give our views on the topic with respect to the data collected.
  2. There seems to be a significant number of very long paragraphs in this paper. Particularly in the Pharmacological treatment section. This makes the material challenging to read and digest, especially when there are multiple studies combined into a single long paragraph. The authors should revisit some of these long paragraphs and break them up to improve readability and clarity of information.
    1. We have tried to be as inclusive as possible regarding the data in the literature, trying to find the right balance with readability.
  3. Lines (L) 26-28 – list of research methods used in this study should be plural – only the first one in the list is currently plural.
    1. We have corrected the sentence as indicated.
  4. L 28 – period should come after the reference
    1. The number in this case refers to a numerical list, not a reference.
  5. L 29 – should “are” be “were”?
    1. We have corrected the word as indicated.
  6. L 46-49 – should all disorders in this sentence be capitalized?
    1. We have corrected the sentence as indicated.
  7. L 49 – should “is” be “has” (past tense- study completed)?
    1. We have corrected the word as indicated.
  8. L 50 should “outpatients’” be “outpatient”?
    1. We have corrected the word as indicated.
  9. L 55 “leaded” should be “lead”.
    1. We have corrected the word as “led”.
  10. L 62 – sentence starting “The majority…” is awkward.
    1. We have corrected as “Most…”.
  11. L 76 – “is” should be “was”
    1. We have corrected the word as indicated.
  12. L 93 – the word “initially” in the sentence is a bit confusing as it suggests other methodology was later used and this was not the case.
    1. We removed the word “initially” as indicated.
  13. L 101 – “is” should be “was”? Does this sentence require any further explanation?
    1. We have corrected the word as indicated.
  14. L 139 – should read “because they were not pertinent..”
    1. We have corrected the sentence as indicated.
  15. L 141 – should read “because they were not focused...”
    1. We have corrected the sentence as indicated.
  16. L 157 – should read “findings”
    1. We have corrected the word as indicated.
  17. L 163 – should read “methods.”
    1. We have corrected the word as indicated.
  18. Table 1. – the authors should consider whether the inclusion a key of abbreviated terms is necessary.
    1. We have included a legend at the end of the table of all abbreviations used.
  19. L 177 – sentence starting “We found… “ the types of methods should be plural – e.g. chart reviews
    1. We have corrected the sentence as indicated.
  20. L 200 – the authors use the term inhomogeneous but the term heterogeneous might be preferable as it is more common term
    1. We have corrected the word as indicated.
  21. L 220 – Should “emergence agitation” be “emergent”?
    1. We have corrected the word as “emergency”.
  22. L 253 – “involved” might be a better word than “associated”.
    1. We have corrected the word as indicated.
  23. L 257-258- Start of the sentence “According to our studies selection” – is awkward – please reword.
    1. We have corrected the sentence as “According to the data collected”.
  24. L 274 – “vomit” should read “vomiting”
    1. We have corrected the word as indicated.
  25. L 277 – I am a little confused by the use of the phrase “ good compliance” in this context as the treatment was i.v., in an emergency clinical setting, with minors and thus it does not seem like at setting in which compliance would be an issue.
    1. We have corrected the sentence as “…improved treatment compliance”.
  26. L 315 – this sentence might read better “ .. a specific drug was problematic because patients were treated with multiple drugs, thus limiting the interpretation of the results of the study.”
    1. We have corrected the sentence as indicated.
  27. L 324 – “scores” should read “score”
    1. We have corrected the word as indicated.
  28. L 337 – “relatively” may be better than “partially”
    1. We have corrected the word as indicated.
  29. L 360 – “medication” should read “medications”
    1. We have corrected the word as indicated.
  30. L 393 – “6- years” should read “6-year”
    1. We have corrected the word as indicated.
  31. L 415-416 – The beginning of the sentence “It was recruited a sample” is awkward
    1. We have corrected the sentence as “A sample … was recruited”.
  32. L 426 – add a period after “lisping.”
    1. We have added the period as indicated.
  33. L 439 – should read “were treated with”
    1. We have corrected the sentence as indicated.
  34. L 442 – should read “reinforced the need”
    1. We have corrected the word as indicated.
  35. L 449 – write out a number when it begins a sentence
    1. We have corrected the word as indicated.
  36. L 457 – “[46]” is the reference number necessary here?
    1. Removed previous repeated bibliographic reference. Reference number [35] remains correct.
  37. L 463 – “finding” should read “findings”
    1. We have corrected the word as indicated.
  38. L 465 – “prevented to make” is awkward
    1. We have corrected the sentence as “no definitive conclusion … could be made because of …”.
  39. L 512 – Should read “One hundred..”
    1. We have corrected the word as indicated.
  40. L 537 and L 666 – As noted above the authors often write from a first-person perspective, but here they also use first person (we) to refer to the scientific field which is somewhat awkward
    1. We have corrected the sentences as indicated.
  41. L 559 – I am not sure about the necessity of the word “consecutive” here
    1. We have removed the word as indicated.
  42. L 607-609 – Across these lines a 4-year old boy subsequently described as an infant – the term “young child” would be more accurate
    1. We have corrected the word as indicated.
  43. L 628 – “shots” might be replaced with “episodes”
    1. We have corrected the word as indicated.
  44. L 448 – “BPI”? Do the authors mean BD-I?
    1. We have corrected the word as indicated in line 648.
  45. L 652- delete word “already”?
    1. We have removed the word as indicated.
  46. L 659 – spelling of pediatric
    1. We have corrected the word as indicated.
  47. L 671 – “drug” should be “drugs”
    1. We have corrected the word as indicated.
  48. L 674 – replace “grade” with “degree”?
    1. We have corrected the word as indicated.
  49. The Discussion section and the beginning of the Summary of evidence section are identical, this repetition seems unnecessary.
    1. We have removed the repetition as indicated.
  50. L 713 – “AP” should be “PA”
    1. We have corrected the word as indicated.
  51. L 718 – delete “the” before ethical issues
    1. We have removed the word as indicated.
  52. L 722 - term “this aspect” should be clarified
    1. We have corrected the sentence as “the safety profile of medications”.
